# The Trend of HIV/AIDS Incidence and Risks Associated with Age, Period, and Birth Cohort in Four Central African Countries

**DOI:** 10.3390/ijerph18052564

**Published:** 2021-03-04

**Authors:** Nodjimadji Tamlengar Martial, Sumaira Mubarik, Chuanhua Yu

**Affiliations:** Department of Epidemiology and Biostatistics, School of Health Sciences, Wuhan University, Wuhan 430071, China; tammartial@whu.edu.cn (N.T.M.); Sumairaawan86@gmail.com (S.M.)

**Keywords:** HIV/AIDS incidence, Cameroon, Chad, Central African Republic, the Democratic Republic of the Congo, APC model

## Abstract

The HIV/AIDS incidence rates have decreased in African countries although the rates are still high in Sub-Saharan Africa. Our study aimed to examine the long-term trend of the overall HIV/AIDS incidence rates in four countries of the central region of Africa, using data from the Global Burden of Diseases (GBD) 2019 study. The Age–Period–Cohort statistical model analysis was used to measure the trends of HIV/AIDS incidence rates in each of the four countries. HIV/AIDS incidence rates decreased slowly in Cameroon (CAM), Chad, and Central African Republic (CAR), but considerably in the Democratic Republic of the Congo (DRC) from 1990–2019. HIV/AIDS incidence rates in the four countries were at their peaks in the age group of 25–29 years. According to the age relative risks, individuals aged between 15 and 49 years old are at high risk of HIV/AIDS incidence in the four countries. The period and cohort relative risks have decreased in all four countries. Although CAM recorded an increase of 59.6% in the period relative risks (RRs) between 1990 and 1999, HIV/AIDS incidence has decreased dramatically in all four countries, especially after 2000. The decrease of the period RRs (relative risk) by nearly 20.6-folds and the decrease of the cohort RRs from 147.65 to almost 0.0034 in the DRC made it the country with the most significant decrease of the period and cohort RRs compared to the rest. HIV/AIDS incidence rates are decreasing in each of the four countries. Our study findings could provide solid ground for policymakers to promptly decrease HIV/AIDS incidence by strengthening the prevention policies to eliminate the public health threat of HIV/AIDS by 2030 as one of the targets of the Sustainable Development Goals (SDGs).

## 1. Introduction

The human immunodeficiency virus (HIV) is known as the virus that affects the immune system, causing a disease called the acquired immunodeficiency syndrome (AIDS) [1]. Until today, HIV/AIDS remains a major public health issue worldwide. From its beginning, it has caused the death of almost 32 million people, and over 74 million people have been infected, with the largest number recorded in low and middle-income countries [2]. The African continent is the most affected by the burden of HIV/AIDS in the world compared to other continents [2,3], and during 2018, over one million new cases were recorded in Africa [3].

In 2019, the Central and the West African countries registered almost five million people who were living with HIV/AIDS [4]. The Republic of Cameroon (CAM), the Central African Republic (CAR), the Republic of Chad (Chad), and the Democratic Republic of the Congo are four countries amongst the eight that constitute the central region of Africa [5]. The estimated numbers of new cases of HIV/AIDS for those four countries were between 4900 to 23,000 during 2019 [6,7,8,9]. AIDS, which is caused by HIV/AIDS, was ranked as the second major cause of death in the African continent, although there has been an elevated practice of education about its prevention and treatment [10]. The percentage of HIV/AIDS incidence in adults aged between 15–49 years old was nearly equal to 3.2% in CAM, 3.6% in CAR, 1.2% in Chad, and 0.8% in DRC in 2019 [11,12,13,14].

Many studies on HIV/AIDS and AIDS-related diseases in that region of Africa have focused generally on the incidence, mortality, and prevalence amongst the populations aged between 15 and 49 years old. However, there is no study revealing the association of age, period, and cohort relative risks to HIV/AIDS incidence in those four countries. The relative risks induced by age, period, and cohort are still unclear in CAM, CAR, Chad, and DRC. Therefore, our study aims to address and elucidate these limitations through comprehensive analysis to evaluate the impact of age, period, and birth cohort relative risks on HIV/AIDS incidence in the respective countries by examining the long-term trend of HIV/AIDS. The national level of estimates and the patterns of HIV/AIDS incidence rates might help to assess the impact of HIV/AIDS prevention policies and re-think strategies to improve the outcome of HIV/AIDS prevention policies.

## 2. Materials and Methods

### 2.1. Data Source

All HIV/AIDS incidence estimates from 1990–2019 and the population data used in this study were extracted from the Global Burden of Diseases (GBD) 2019 for all four countries [15]. Estimates such as incidence, death, prevalence, years of life lost (YLL), year lived with disability (YLD), and disability-adjusted life years (DALYs) are regrouped by age group, sex, year, and location on the GBD. The GBD database is controlled by the Institute for Health Metrics and Evaluation (IHME) and is based in Seattle, Washington State, USA. Generally, household surveys with complete summary birth histories, censuses, vital registration, disease surveillance system, and sample registration systems constitute the primary data input for the GBD. The data used for our analysis are located in the GBD data input in the global health data exchange (GHDx) section result tools. The dataset obtained from GBD was in rates (per 100,000). Ethical approval was not needed for this study because there was no direct involvement of human subjects.

### 2.2. Statistical Analysis

HIV/AIDS incidence rates were evaluated through an age–period–cohort model analysis. The age effects represent different risks during different periods of life. Period effects indicate the population-large exposure at a specific point of time, and different risks in different birth cohorts are mainly reflected by cohort effects [16,17]. To isolate the different contributions of age, period, and cohort, we decomposed the age and cycled a queue into their linear and nonlinear constituents [18]. This decomposition also produced many essential functions such as net drift, local drifts, longitudinal age curve, period, and cohort deviations [19]. The local drifts represent the annual percentage changes in each age group, and the net drift represents the overall annual percentage changes of the adjusted age group over time. The longitudinal age curve is adjusted for period deviations and represents the fitted longitudinal age-specific rates in the reference cohort. The expected age-specific rates in the reference period adjusted for cohort effects are expressed by the cross-sectional age curve. The period relative risk (RR) defines the period RR adjusted for age and nonlinear cohort contributions of each period relative to the reference period. The cohort relative risk (RR) defines the cohort RR adjusted for age and nonlinear period contributions in each cohort relative to the reference period [20]. Incidence rates and demographic statistics were decomposed into five consecutive years before conducting our age–period–cohort (APC) analysis. They were arranged from 1990–1994 (median 1992.5) to 2015–2019 (median 2017.5). Consecutive five-year age groups were set from 15–19 (median 17.5) to 75–79 (median 77.5). The birth cohort arrangements concerned those born from 1975 to 2004 and were also divided into five consecutive years for the APC analysis. The reference values were selected as the lower two central values in the event of an even number of categories. Wald chi-square tests were conducted with a *p*-value set at *p* ˂ 0.01 for statistical significance. Despite the APC analysis advantages, APC possesses limitations such as the uncertainty principle and identifiability problems [20]. The uncertainty principle indicates the measurements of absolute rates in cohorts that are not frequently taken into consideration by most epidemiological cohort and case-control researchers [20]. The identifiability problem is the fact that the three scales of age, period, and cohort are collinear from the equation cohort equals period minus age; thus, the log-linear trends in the rates cannot only be representing the contributions of age, period, and cohort [17].

We used the APC Web Tool (Biostatistics Branch, National Cancer Institute, Bethesda, MD, USA) software in our statistical analysis, and RStudio version 1.3.959 2009–2020 to realize our graphics.

## 3. Results

### 3.1. Trends of HIV/AIDS Incidence in Cameroon, Central African Republic, Chad, and the Democratic Republic of the Congo

The trends of crude and age-standardized HIV/AIDS incidence rates in the four countries displayed similar declining patterns (Figure 1). In 1990, the CAR had the highest HIV/AIDS incidence rate compared to the rest of the countries. From 1990–2000, CAM and Chad recorded an upward pattern in their HIV/AIDS incidence rates before decreasing until 2019. Although HIV/AIDS incidence rates have decreased in all four countries, it remains higher in the CAR compared to CAM, Chad, and the DRC with the DRC having the lowest incidence rate in 2019, as represented in Figure 1.

### 3.2. Net Drifts and Local Drifts of HIV/AIDS Incidence in Cameroon, Central African Republic, Chad, and the Democratic Republic of the Congo

Table 1 shows the annual percentage change in each age group defined as the local drifts and the overall annual percentage changes of the adjusted age group over time, which is the net drift of HIV/AIDS incidence rates in the four countries. All local drifts and net drifts values were represented by negative values. The net drift value was around −6.74% (95% CI: −6.91%, −6.57%) in CAM, −8.16% (−8.30%, −8.02%) in the CAR, −5.63% (−5.87%, −5.38%) in Chad, and around −11.34% (−11.69%, −10.99%) in the DRC. However, HIV/AIDS incidence annual percentage changes (local drifts) in each age group had negative values, indicating a decrease. Some age groups recorded upward behavior in each of the countries (Table 1). The age groups of 55–69 years in CAM, 60–74 years in the CAR, 70–79 years in Chad, and 25–34 and 60–79 years old in the DRC have seen their annual percentage changes elevate slightly compared to each respective previous age group. When looking at all four countries, HIV/AIDS incidence rates decreased the most in the adjusted age group of the DRC over time.

### 3.3. HIV/AIDS Incidence and Age Relative Risks

Figure 2 displays the HIV/AIDS incidence rates in each age group over the years (1990–2019). We considered the lowest year between two intervals for each period (for example, 1990 represents 1990–1994, and 2015 for 2015–2019). According to all age groups, all four countries displayed decreasing trends. CAR and DRC displayed similar patterns, and CAM and Chad showed a steady or slight upward pattern in all age groups from 1990–1999 before declining until 2019. In all four countries, HIV/AIDS incidence rates were at their peaks in the age groups of 25–29 years old.

Figure 3 describes the age relative risks or the age risk ratios (RRs) of the HIV/AIDS incidence rates of all four countries in the population aged between 15–79 years old. All trends displayed analogous patterns, and CAM and Chad had approximately the same trend line. Compared to the threshold value (value equals 1) of the age RRs, the population younger between 15–49 years are at high risk of HIV/AIDS incidence in CAM, the CAR, Chad, and the DRC. In all four countries, the age RRs of HIV/AIDS incidence decreased rapidly in the age group (15–49 years) at high risk compared to the rest of the age categories (50–79 years).

### 3.4. HIV/AIDS Incidence and Period Relative Risks

Figure 4 displays the trends of the period relative risks (RRs) of HIV/AIDS incidence rate in CAM, the CAR, Chad, and the DRC from 1990 to 2019. Between 1990 and 1999, the period RRs of HIV/AIDS incidence in CAM increased by almost 59.6% before declining. After 2000, the period RRs pf HIV/AIDS incidence dramatically decreased in all four countries. Between the study period of 1990–2019, the trends of period RRs of the DRC and CAR displayed synonymous patterns. All four countries recorded a decrease in period RRs from 1990 to 2019, and the decrease of period RRs was nearly 4.6 times in CAM, 9.8 times in the CAR, four times in Chad, and 20.6 times in the DRC during 2019, compared to 1990.

### 3.5. HIV/AIDS Incidence and Cohort Relative Risks

The cohort RRs of HIV/AIDS incidence in CAM, the CAR, Chad, and the DRC are represented in Figure 5. In the cohort born before 1955, the DRC had the highest and Chad had the lowest cohort RRs compared to the other countries. All four trends of the cohort RRs displayed a similar declining pattern with a rapid decrease before 1955, and a slower decrease after 1955. The cohort RRs decreased from 13.17 to 0.02 in CAM, from 28.49 to 0.01 in the CAR, from 7.43 to 0.04 in Chad, and finally from 147.65 to 0.00 in the DRC. All cohort RRs of HIV/AIDS incidence had consistently decreased to nearly zero in all four countries from the first cohort group (1915–1919) to the last cohort group (2000–2004).

The Wald tests demonstrated statistical significance for the net drifts, the local drifts, the age RRs, the period RRs, and the cohort RRs at *p* ˂ 0.01.

## 4. Discussion

In our study, we investigated the long-term trend of HIV/AIDS incidence from 1990 to 2019 in four countries of the central region of the African continent using an age, period, and cohort statistical model. Compared to existing researches, our study is the first to examine the association of age, period, and cohort relative risks to HIV/AIDS incidence in Cameroon (CAM), the Central African Republic (CAR), Chad, and the Democratic Republic of the Congo (DRC) from 1990 to 2019. Our findings revealed that HIV/AIDS incidence decreased in CAM, the CAR, Chad, and the DRC from 1990 to 2019. However, HIV/AIDS incidence rates in CAR during 1990 and 2019 were the highest compared to the other countries, and between 1990 and 2000, the HIV/AIDS incidence rates in CAM rose before declining until 2019. In CAM, the first national strategic plan was drafted from 2000 to 2005 to fight the increase in HIV/AIDS incidence in the country [21]. Therefore, HIV/AIDS incidence in CAM started declining considerably and continuously after 2000, based on our findings. Although strategic policies were elaborated in all four countries and they all contributed to the decline in the HIV/AIDS incidence rate in their populations, respectively [22,23,24,25], the outcomes in real life might not be equally perceived in all four countries. This could be the reason why the DRC had the lowest HIV/AIDS incidence rate in 2019 compared to the others, as shown in our research. The efforts made by governmental and non-governmental institutions played a significant role in the decrease of HIV/AIDS incidence rates from 1990 to 2019 in those four countries of the central region of Africa, although this decrease is not yet sufficient because acquired immune deficiency syndrome (AIDS), which is caused by the HIV/AIDS, remains the second leading cause of death on the continent [10].

The net drift, which represents the overall annual percentage changes of the adjusted age group over time, and the local drifts, which are the annual percentage change in each age group, were all below zero in all four countries. According to the net drift value of each country, HIV/AIDS incidence of the adjusted age group decreased the most overtime in the DRC compared to CAM, the CAR, and Chad. Although all countries had negative values for the annual percentage changes in each age group, some age categories recorded upward behavior. The age groups of 55–69 years in CAM, 60–74 years in the CAR, 70–79 years in Chad, and 25–34 and 60–79 years old in the DRC have seen their annual percentage changes elevating slightly compared to each respective previous age group. This finding might be because of the following facts. On one hand, all four countries had a similar socio-cultural background mainly because of their geographical location. Sexual practice with multiple partners is common in CAM, the CAR, Chad, and the DRC [26]. Socio-demographic factors such as polygamous households have been noticed in those countries [27,28]. On the other hand, young adults are pretty likely to have the advantages of health and prevention and treatments, and are exposed to late diagnosis because many live with HIV/AIDS without being tested due to the taboo behind HIV/AIDS [29]. Consequently, the slight elevation in the annual percentage changes in the elderly group might be associated with late diagnosis, lack of sexual education, and health care [27,28,29]. Governmental and non-governmental organizations involved in the fight for the prevention of HIV/AIDS should give particular attention to the age groups associated with the upward behavior of the HIV/AIDS incidence rate in the populations of the four countries.

HIV/AIDS incidence rates were at the peak in the age group of 25–29 years old in all four countries. According to the age relative risks (RRs) associated with HIV/AIDS incidence, the population aged between 15–49 years old in the CAR, Chad, the DRC, and CAM are at higher risk of HIV/AIDS incidence. The association of individuals aged from 15–49 years old to the risk of HIV/AIDS incidence had been discussed in other studies [30,31]. The HIV/AIDS prevalence rates among adults aged between 15–49 years were the highest in Africa compared to other continents in 2019 [32]; except for the DRC, HIV/AIDS prevalence in the same age category (15–49 years) in 2019 remained higher in CAM, the CAR, and Chad compared to 1990 [33]. Alcohol consumption, drug usage, and condom-free sexual stimulations are generally associated with young adults around this age category [34]. These findings might explain the relative risks of HIV/AIDS incidence in this age category. Furthermore, homosexuality had been a taboo in Africa for centuries. However, this sexual behavior has been adopted by many individuals in the world, and “same-sex marriage” has been approved in some countries including one African country officially [35]. In 2011, a study demonstrated that more “sexually-straight” youth engaged in same-sex activities [36], and that men who sold sex or had sex with men were marked as a population at high risk of HIV/AIDS incidence, especially in low- and middle-income countries [37,38]. HIV/AIDS incidence rates are increasing in the population of men who have sex with men, injectable drug users, and sex workers worldwide, and especially in the north of Africa and the Middle East [39,40]. Therefore, governmental and non-governmental institutions should begin to make adapted policies and promote sexual education programs according to the evolution of society globally. Medical health care providers should be trained to educate the population about the risks during unprotected intercourse in heterosexuals and homosexuals including sex workers. The lack of knowledge about HIV/AIDS prevention techniques, especially in the homosexual community, might play a major function behind these findings. Condoms should be more available and accessible to all social class structures. The use of HIV/AIDS self-testing tools should be encouraged, and HIV/AIDS prophylaxis therapy education should be offered regularly to the populations at risk. Understanding the sexual behavior of the population might considerably impact the decrease in the HIV/AIDS incidence risks in the 15–49 years age category. Policies against gender-based and sexual violence should also be strengthened in each of the four countries.

The period and cohort RRs of HIV/AIDS incidence of the four countries all displayed decreasing patterns in trends. The trend of period RRs in CAM showed an upward behavior from 1990 to 1999 before decreasing until 2019, and the period RRs in the DRC decreased the most compared to the other countries (nearly 20.6-folds from 1990 to 2019). The cohort RRs in CAM, the CAR, Chad, and the DRC have all decreased significantly to nearly zero from the cohort born in 1915 to 2004. In the cohort born before 1955, the cohort RRs decreased dramatically compared to the cohort born after 1955. The fact that period effects can have some influence on some age groups and people from different birth cohort during separated years can also influence the period association, their separate interpretation tends to be relatively difficult [20]. The upward behavior recorded in the period RRs of HIV/AIDS incidence in CAM from 1990–1999 could be justified by the fact that the country drafted its first national program to fight against the HIV/AIDS disease during 2000 [21]. Therefore, the decreasing trends in period and cohort RRs in the four countries are mainly associated with efforts conducted by governmental and non-governmental institutions to decrease the HIV/AIDS burden in those respective countries [22,23,24,25]. The most significant decrease of the period and cohort RRs in the DRC might also be explained by socio-cultural and demographic factors, which have influenced the population of DRC and their sexual behavior [41,42,43]. Although governmental and non-governmental institutions have made many efforts in decreasing the HIV/AIDS burden in the respective countries, the decrease is not yet sufficient [10]. The four countries also had a resemblant socioeconomic background. The human development index (HDI) is a statistical index generated by the assessment of life expectancy at birth, education coverage, and the gross national income per capita [44]. This statistical variable indicates the human development coverage in different domains such as long healthy life, being well-educated and informed, and having proper living standards [42]. In CAM, the CAR, Chad, and the DRC, the HDI values were between 0.397 to 0.563, with CAM and CAR having the highest and the lowest values, respectively [45]. Such low values indicate that poverty, low literacy rate, and improper health care and living conditions are not improving for the population of the four countries. Consequently, national and international organizations should join hands and fight strongly against corruption to alleviate poverty and make valuable policies and strategies to improve the outcome of constituents of the human development index in those four countries, respectively, because HIV/AIDS incidence is increasing faster in marginalized nucleus category in Sub-Saharan countries recently [46]. This new face of the epidemic might require to re-think strategies used in the fight against HIV/AIDS burden to reach the sustainable development goals set to end the epidemic as a public health threat by 2030.

Our study had some limitations. First, our research did not discuss any subgroup of HIV/AIDS or gender specificity. The second constraint is related to the age–period–cohort analysis (identifiability and uncertainty principle). These parameters could not be avoided because the interpretation of results from an individual’s level does not necessarily equate to the interpretation at the population level. However, many other studies have been conducted using the web tool APC model as we did in our study [34,47,48]. Therefore, in the future, we will perform a large-scale study to validate the hypothesis associated with our present study.

## 5. Conclusions

In summary, although HIV/AIDS incidence has decreased over the last three decades in Cameroon, the Central African Republic, Chad, and the Democratic Republic of the Congo, the decrease is not sufficient because the disease called the acquired immunodeficiency syndrome (AIDS) caused by HIV/AIDS remains one of the leading causes of death in all four countries. Young adults aged from 15 to 49 years are at high risk of HIV/AIDS incidence according to our analysis. More strategic policies regarding public health, the health care system, and poverty alleviation should be implied by national and international organizations to control the sexual behavior of this age category. Massive campaigns about sexual education for health care practitioners and the populations, especially about HIV/AIDS prophylaxis, should be organized on a regular basis. This approach could help to significantly decrease the HIV/AIDS incidence in Cameroon, Chad, the Central African Republic, and the Democratic Republic of the Congo.

## Figures and Tables

**Figure 1 ijerph-18-02564-f001:**
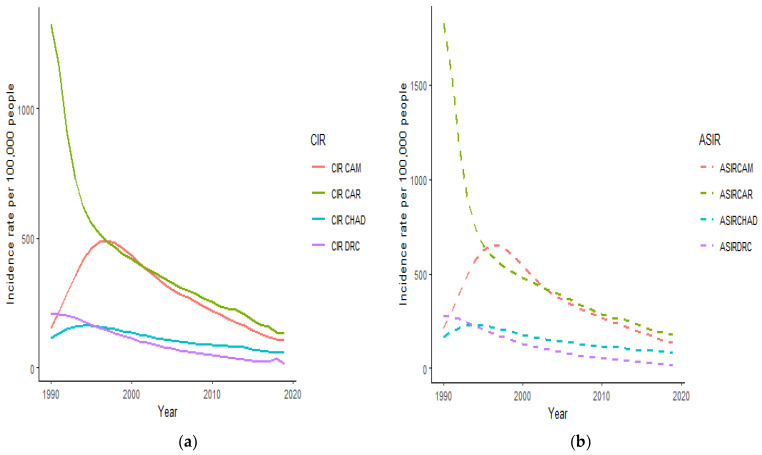
Crude incidence rate (CIR) (**a**) and age-standardized incidence rate (ASIR) (**b**) of HIV/AIDS infection incidence in Cameroon (CAM), Central African Republic (CAR), Chad, and the Democratic Republic of the Congo from 1990 to 2019.

**Figure 2 ijerph-18-02564-f002:**
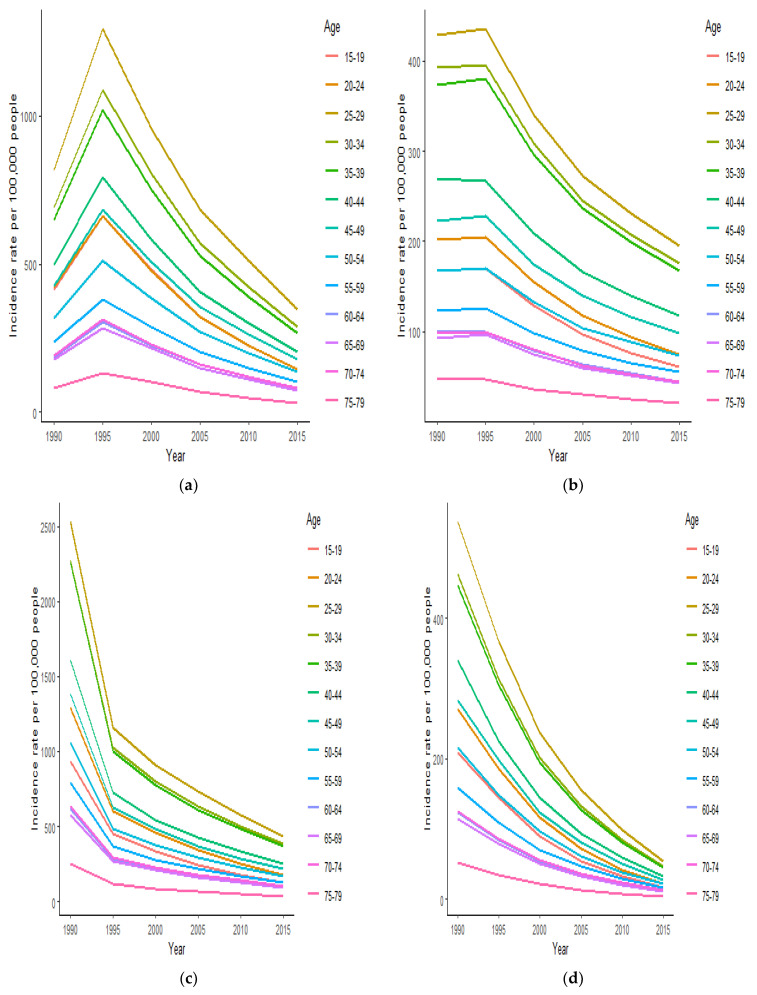
HIV/AIDS incidence rates in each age group in Cameroon (CAM) (**a**), Chad (**b**), the Central African Republic (**c**), and the Democratic Republic of the Congo (**d**).

**Figure 3 ijerph-18-02564-f003:**
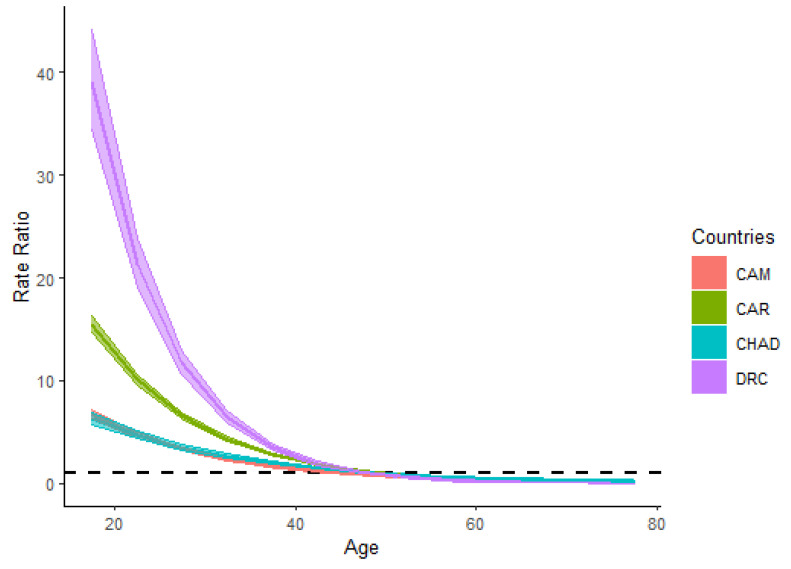
Age relative risks (Age RRs) of HIV/AIDS incidence and their 95% confidence intervals in Cameroon (CAM), Central African Republic (CAR), Chad, and the Democratic Republic of the Congo.

**Figure 4 ijerph-18-02564-f004:**
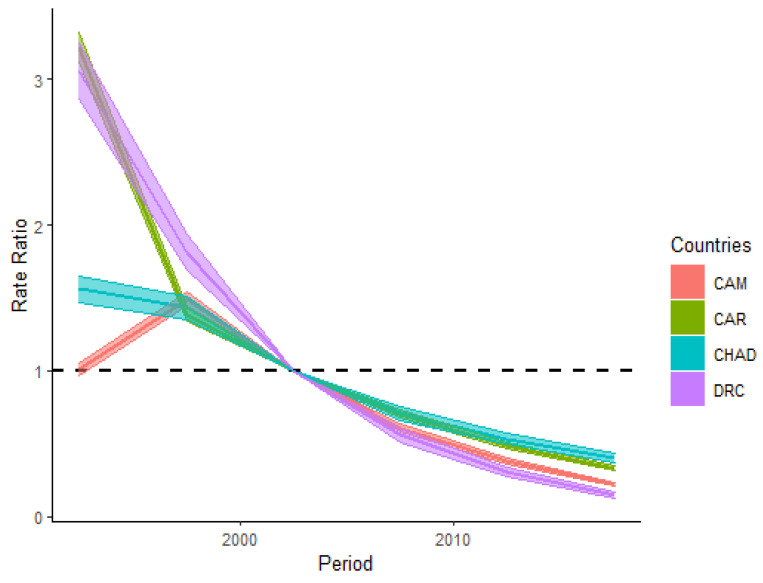
Period relative risks (period RRs) of HIV/AIDS incidence and their 95% confidence intervals in Cameroon (CAM), the Central African Republic (CAR), Chad, and the Democratic Republic of the Congo (DRC).

**Figure 5 ijerph-18-02564-f005:**
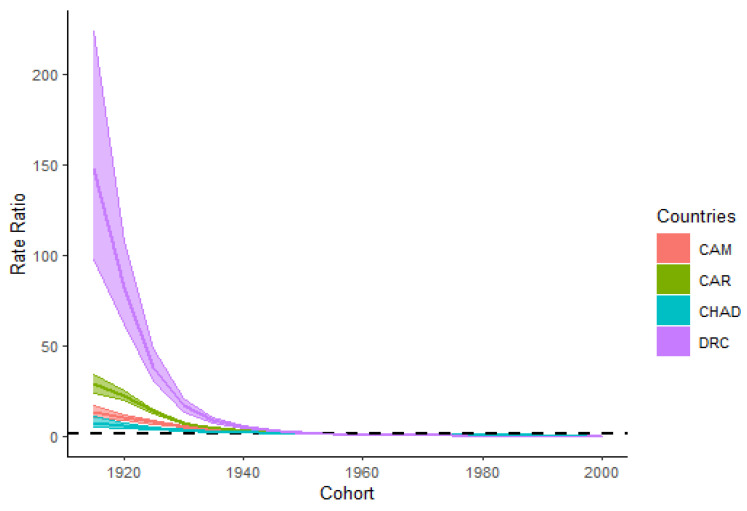
Cohort relative risks (cohort RRs) HIV/AIDS incidence and their 95% confidence intervals in Cameroon (CAM), the Central African Republic (CAR), Chad, and the Democratic Republic of the Congo (DRC).

**Table 1 ijerph-18-02564-t001:** Local drifts and Net drifts of HIV/AIDS incidence in the Central African Republic (CAR), Cameroon (CAM), Chad, and the Democratic Republic of the Congo (DRC).

Age	CAR		CAM		CHAD		DRC	
	Local Drift (%)	95% CI	Local Drift (%)	95% CI	Local Drift (%)	95% CI	Local Drift (%)	95% CI
15–19	−9.47	−9.99; −8.95	−8.44	−8.94; −7.93	−7.23	−8.02; −6.42	−11.96	−13.28; −10.63
20–24	−9.14 ꜜ	−9.46; −8.82	−8.05 ꜜ	−8.39; −7.72	−6.76 ꜜ	−7.29; −6.24	−11.70 ꜜ	−12.47; −10.92
25–29	−8.93 ꜜ	−9.15; −8.71	−7.64 ꜜ	−7.89; −7.39	−6.36 ꜜ	−6.73; −5.99	−11.75 ꜛ	−12.28; −11.23
30–34	−8.83 ꜜ	−9.03; −8.64	−7.30 ꜜ	−7.54; −7.07	−6.24 ꜜ	−6.57; −5.90	−11.94 ꜛ	−12.40; −11.47
35–39	−8.57ꜜ	−8.77; −8.37	−6.94 ꜜ	−7.18; −6.71	−6.2 ꜜ	−6.54; −5.85	−11.85 ꜜ	−12.31; −11.39
40–44	−8.03 ꜜ	−8.24; −7.82	−6.52 ꜜ	−6.78; −6.26	−6.08 ꜜ	−6.45; −5.71	−11.45 ꜜ	−11.93; −10.97
45–49	−7.41 ꜜ	−7.65; −7.19	−6.11 ꜜ	−6.39; −5.83	−5.86 ꜜ	−6.27; −5.45	−10.79 ꜜ	−11.31; −10.26
50–54	−7.02 ꜜ	−7.27; −6.77	−5.95 ꜜ	−6.26; −5.64	−5.45 ꜜ	−5.91; −4.99	−10.21 ꜜ	−10.79; −9.64
55–59	−7.00 ꜜ	−7.28; −6.72	−6.15 ꜛ	−6.50; −5.80	−4.87 ꜜ	−5.39; −4.35	−10.13 ꜜ	−10.77; −9.49
60–64	−7.37 ꜛ	−7.67; −7.06	−6.45 ꜛ	−6.83; −6.07	−4.50 ꜜ	−5.07; −3.92	−10.55 ꜛ	−11.24; −9.85
65–69	−8.48 ꜛ	−8.81; −8.15	−6.75 ꜛ	−7.17; −6.34	−4.33 ꜜ	−4.96; −3.70	−11.57 ꜛ	−12.32; −10.81
70–74	−9.35 ꜛ	−9.71; −8.99	−6.67 ꜜ	−7.16; −6.18	−4.49 ꜛ	−5.22; −3.76	−12.66 ꜛ	−13.48; −11.83
75–79	−9.29 ꜜ	−9.82; −8.77	−6.37 ꜜ	−7.14; −5.60	−4.88 ꜛ	−5.95; −3.80	−13.11 ꜛ	−14.25; −11.95
Net drift (%)	−8.16	−8.02; −8.30	−6.74	−6.57; −6.91	−5.63	−5.38; −5.87	−11.33	−10.98; −11.68

Note: 95% CI, 95% confidence interval; ꜛ, upward values; ꜜ, downwards values; %, percentage.

## Data Availability

Data presented in this study are openly available from the Global Burden of Disease. Available online: http://ghdx.healthdata.org/gbd-results-tool (accessed on 5 November 2020).

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
