# Peer review of "The Trend of HIV/AIDS Incidence and Risks Associated with Age, Period, and Birth Cohort in Four Central African Countries"

_ijerph, 2021, doi:10.3390/ijerph18052564_

Round 1
Reviewer 1 Report
I commend the authors for their study. I have a few comments.
Page 2: lines 62-63: “The GBD database is controlled by the Institute for Health Metrics and Evaluation 62 (IHME) and is based in Washington, D.C. in the USA.’’ This reviewer would like to note that IHME is based in Seattle, Washington State, USA, not Washington, D.C. in the USA.
Page 7.lines 196-197. “ An online report demonstrated that the populations of CAM, CAR, Chad, and DRC have more than one sexual partner [28]. I suggest that this statement be changed to, Sexual practice with multiple partners is common in CAM, CAR, Chad, and DRC [28].
Page 9. Line 169. “Young adults aged under 49 years old are at risk of HIV incidence according to our analysis.” I suggest that throughout the manuscript a range of age categories such as 15-49 be stated, instead of under 49, which of course would include pediatric age groups.
The reference formatting is not consistent. For example, in the first reference, the year follows the author(s). In the last reference, the year comes last. Moreover, I suggest that the terms accessed on the date, or retrieved from be used as appropriate just before the URLs from which the references are obtained.
Author Response
Dear Editor
Please see the attachment to this email for the response to reviewer 1 comments.
Thank you

Reviewer 2 Report
Thanks for allowing me to read your valuable work. I will list my comments by section for you to consider.
- Abstract
- Usually the HIV incidence is reported separately from the AIDS incidence, please define in the background why you combined them, what the combined HIV/AIDS incidence means in the case of Western Africa.
- When was an increase in the HIV/AIDS incidence observed over the 20 year time period for the above 40 year olds.
- What does this mean: "
-
age relative risks were instead extended to the population below 49"
-
- I would also say the HIV incidence rates have decreased dramatically especially since 2000.
- Keywords are highly repetitive, should be AIDS, cohort, Cameroon, Chad, Central African Republic, Democratic Republic of Congo, period
Background
Line 34 it is has killed and over 74 million have been
Line 43 what year does this refer to?
Line 45 I would suggest instead" was an elevated practice of education about its prevention and treatment, rather despite high levels of prevention efforts and treatment implementation"
Line 48 Please seperate HIV from AIDS. On refers to the number of infections, the other refers to the people who develop AIDS based a a set of specific indicators.
Line 50-51 Why is this analysis important?
Line 63 the Institute for Health Metrics is in Seattle and not in Washington DC.
Line 61 Please spell out YYL, YLD and DALY I seem to not find them again in the result section.
Line 66 What is a GHDx tool? Please describe the data better so we understand what you did.
Line 96 Which National Cancer Institute are you referring too?
Results
Figure 2 the text to Figure 2 seems to say the opposite than what is displayed in the graph. I am not an expert in this method but the text need to improve to better explain what is shown. What is the difference between the two types of drifts?
Does Figure 2 and Figure 3 not contradict each other? How can the drift go up for 60-69 year olds and the RR decrease by the age of 40? So according to Figure 3 people age 20-44 are at highest risk to get infected and develop AIDS. Is that what you are saying?
Line 123 You are also jumping between people younger than 49 and the a sentence later younger than 44. Please be consistent with your numbers.
Figure 5 I am confused by as well. How could someone who was 80 years old be at the highest risk for HIV infection. Please explain this graph better.
Discussion
Please delete line 176-177. The add nothing to the discussion.
Line 180-181 how do you support this statement. Based on your data?
Line 200 is an unsupported speculation, please provide a reference.
Line 203 how do you get to these age numbers?? Not from any of the graphs, past 40 you have a straight line.
Lines 209-225 introduces new concepts such as homosexuality that have never been mentioned before, and I think you do not have data to support such statements.
What I am missing is how does this data connect to data from other parts in Africa
There is no data shown on polygamous house holds or religious beliefs but a lot of interpretations are made in these regards. Please substantiate your conclusions better.
In line 269 you speak now of HIV incidence only? What happened to AIDS?
Author Response
Dear Editor
Please see the attachment for the response to Reviewer 2 comments.
Thank you
Round 2
Reviewer 2 Report
Thanks for your thoughtful revisions.